# Cranial Neural Crest Cells and Their Role in the Pathogenesis of Craniofacial Anomalies and Coronal Craniosynostosis

**DOI:** 10.3390/jdb8030018

**Published:** 2020-09-09

**Authors:** Erica M. Siismets, Nan E. Hatch

**Affiliations:** 1Oral Health Sciences PhD Program, School of Dentistry, University of Michigan, Ann Arbor, MI 48109-1078, USA; esiis@umich.edu; 2Department of Orthodontics and Pediatric Dentistry, School of Dentistry, University of Michigan, Ann Arbor, MI 48109-1078, USA

**Keywords:** craniofacial, skull, craniofacial abnormalities, neural crest, craniosynostoses, growth and development

## Abstract

Craniofacial anomalies are among the most common of birth defects. The pathogenesis of craniofacial anomalies frequently involves defects in the migration, proliferation, and fate of neural crest cells destined for the craniofacial skeleton. Genetic mutations causing deficient cranial neural crest migration and proliferation can result in Treacher Collins syndrome, Pierre Robin sequence, and cleft palate. Defects in post-migratory neural crest cells can result in pre- or post-ossification defects in the developing craniofacial skeleton and craniosynostosis (premature fusion of cranial bones/cranial sutures). The coronal suture is the most frequently fused suture in craniosynostosis syndromes. It exists as a biological boundary between the neural crest-derived frontal bone and paraxial mesoderm-derived parietal bone. The objective of this review is to frame our current understanding of neural crest cells in craniofacial development, craniofacial anomalies, and the pathogenesis of coronal craniosynostosis. We will also discuss novel approaches for advancing our knowledge and developing prevention and/or treatment strategies for craniofacial tissue regeneration and craniosynostosis.

## 1. Introduction

Craniofacial anomalies are among the most common of all birth defects. Like many congenital defects, most craniofacial anomalies occur due to a combination of genetic and environmental factors, the latter including maternal exposure to toxins (including tobacco and alcohol) and certain medications [1,2]. Craniofacial anomalies can occur in isolation (the anomaly occurs with no other defects and with no established genetic basis), as part of an established syndrome with a known constellation of defects due to single gene mutation or chromosomal abnormality, or in association with additional development defects, but without a known genetic basis. The pathogenesis of developmental craniofacial abnormalities commonly involves defects in the migration, proliferation, and fate of cranial neural crest cells and their derivatives. These neural crest developmental abnormalities lead to a variety of developmental defect syndromes, overall referred to as neurocristopathies, including those that are found in cleft palate [3], Treacher Collins syndrome [4], Pierre Robin sequence [5], and craniosynostosis [6]. The coronal cranial suture is prematurely fused in many individuals with syndromic craniosynostosis, and particularly in those with mutations in FGFR2 or Twist [7,8]. Unlike other cranial sutures, the coronal suture develops between mesoderm-derived parietal and neural crest-derived frontal bone rudiments, and it maintains the boundary between these bone tissues of different embryonic origin during growth [9,10]. Therefore, the role of neural crest cell defects in the pathogenesis of coronal craniosynostosis is of particular interest. In this review, we will first provide an overview of neural crest cells in mammalian craniofacial development and cranial neural crest cells in craniofacial anomalies. We will subsequently focus on cranial neural crest cells in mammalian skull development and the pathogenesis of craniosynostosis, with particular emphasis on premature fusion of the coronal suture, due to the large amount of literature that exists in this field. Finally, we will discuss novel approaches for prevention and treatment.

## 2. Neural Crest Cells

Neural crest cells are a transient stem cell population characterized by their strong migratory potential and multipotency. The neural crest is unique to vertebrates and it can be divided into four major sub-populations: cranial, cardiac, vagal, and trunk. Each defined sub-population of neural crest cells differentiates into diverse, specialized cells, and tissues particular to the axial level of origin. Neural crest cells arise in the embryonic ectodermal germ layer at the neural plate border during gastrulation. A precise combination of BMP, FGF, Wnt, and Notch inductive signals from the ectoderm triggers an epithelial-to-mesenchymal transition (EMT) in the neural plate border territory [11] (Figure 1). EMT is a complex, multi-step process required to transform adherent neuroepithelial cells into migratory neural crest cells. This transformation is induced through the dissolution of adherens junctions through cadherin expression changes. EMT is regulated by Wnt/β-catenin signaling and it can be thought of as a spectrum of epithelial and mesenchymal cell fates, rather than an exact binary switch [12]. The first cadherin switch allows neural crest cells to delaminate from the neural tube and the second switch grants neural crest cells their migratory potential [13]. The onset of EMT in mouse embryos is controlled by glycogen synthase kinase 3 (GSK3), a component of the β-catenin destruction complex [14,15]. GSK3 phosphorylates Snail and Twist in premigratory neural crest cells (Lander), which along with other neural crest transcription factors *FoxD3*, *Sip1/Zeb2*, and *Sox9/10,* will downregulate epithelial cadherins and cadherin6B [16,17]. Epithelial cadherin signatures are replaced with mesenchymal *cadherin-7* and *cadherin-11*, as neural crest cells become migratory [18]. Metalloproteinases (including MMP-2, -9, -14 and ADAM-10, -13, and -19) are required for neural crest delamination and migration to degrade cadherins within and extracellular matrix proteins around neural crest cells [12,19,20,21]. Defects in components that are involved in EMT can affect craniofacial development. For example, the loss of GSK3 in mouse embryos results in cleft palate [22], reduced frontal bone size, and enlarged frontal fontanelle [23].

It was previously believed that neural crest cells migrate on top of mesoderm cells or that their migration is caused by the dynamics of surrounding ectoderm and mesoderm cells [24]. However, recent in vivo time-lapse imaging in chick embryos revealed that neural crest cells squeeze between ectoderm and mesoderm cells, but do not go over the top of these other cells [25]. In this study, neural crest cells migrated faster than surrounding cells. Mesoderm cells were also shown to be migratory, but these cells migrated slower and were later to emerge [25]. Importantly, this study established that neural crest and mesoderm cells are independent migratory populations, which are influenced by each other’s presence and behavior.

Neural crest cell migration is directed by chemotaxis. Neural crest cells from the hindbrain region migrate to populate the pharyngeal arches, which subsequently form much of the craniofacial skeleton [26]. Neural crest cell migration into the pharyngeal arches involves local cues, particularly those that arise from the ectoderm [27]. In the chick embryo, ectodermal cells induce short-range gradients of VEGF to direct the neural crest migration front [25,28,29]. In the mouse, fibroblast growth factor 2 (FGF2) recruits the migration of fibroblast growth factor receptor (FGFR)-expressing mesencephalic neural crest cells and later promotes the proliferation or differentiation of these cells [30]. FGF8 can promote FGF2 activity and neural crest cell chemotaxis [30,31]. More recent studies indicate that FGF8 can also promote neural crest cell survival in the first pharyngeal arch, as evidenced by data showing that FGF8 inactivation caused massive NCC apoptosis in the first pharyngeal arch [32]. FGF8 expression must be regulated in the craniofacial complex for proper development. Insufficient FGF8 in developing skull bone results in craniosynostosis, and excess FGF8 severely disrupts intramembranous craniofacial ossification [33]. FGF8 expression is also known to control neural crest expansion of maxillary primordia and the frontal bone in craniofacial ciliopathies [34,35]. Hedgehog signaling also promotes neural crest survival in the pharyngeal arches [36] and it is essential for normal facial patterning and growth [37]. Neural crest cells in the pharyngeal arches eventually differentiate to form craniofacial tissue. Non-canonical Notch signaling regulates the proliferation and differentiation of neural crest cells into osteoprogenitors [38,39]. The TGF-beta superfamily regulates neural crest differentiation into a variety of craniofacial structures, including osteoprogenitor cells of the frontal bone rudiment and the palatal mesenchyme [40,41,42,43,44].

It is important to note here that much of our knowledge regarding neural crest development comes from non-mammalian model organisms, including *Xenopus,* chick, and zebrafish [45]. This review is centered on mammalian craniofacial development and anomalies. Mice do differ in some aspects of process and regulation of neural crest development. For example, EMT occurs prior to neural tube closure in the mouse, which is unlike other model organisms [46]. Non-canonical Wnt signaling is required for neural crest cells to gain migratory potential in *Xenopus,* chick, and zebrafish [47,48,49,50], but it is not required in mouse neural crest cells [51]. Instead, canonical Wnt signaling is required for neural crest migration and differentiation in the mouse embryo [52]. We acknowledge that non-mammalian model organisms have significantly contributed to our knowledge of the neural crest; however, the remainder of this review will focus on mammalian neural crest and craniofacial development.

## 3. Cranial Neural Crest Cells in the Pathogenesis of Craniofacial Anomalies

Defects in neural crest cell formation, migration, proliferation, and differentiation can lead to a variety of developmental defects given that neural crest-derived tissues include those that are essential for heart, nervous system, skin pigmentation and craniofacial development. In humans, the term neurocristopathy is used to describe this large and varied group of human syndromes caused by defects in neural crest cell development. Neurocristopathies, including a new classification system, as defined by axial origin of the involved neural crest cell population(s), is well reviewed in a recent publication by Vega-Lopez et al., 2018 [53]. Notably, some neurocristopathies result from defective development of a single neural crest cell population, while others involve defects in more than one neural crest cell population. In this review, we will focus on those that are specific to the cranial neural crest cell population.

Cranial neural crest cell defects underpin many craniofacial anomalies. Treacher Collins syndrome (mandibulofacial dysostosis) is perhaps the first well recognized craniofacial syndrome that occurs due to cranial neural crest cell defects. This autosomal dominant disorder occurs in approximately one in 50,000 live births [54]. Individuals with Treacher Collins syndrome exhibit bilateral periorbital anomalies, including downward slanting of palpebral fissures, maxillary, mandibular, and facial bone hypoplasia, with hairline displacement plus external and middle ear defects due to cranial neural crest cell deficiency [55,56]. The syndrome predominantly occurs due to inactivating mutations in *Tcof1*, the gene for TREACLE, which was originally shown to be essential for the genesis of ribosomes [57]. Earlier studies of *Tcof1*^+/−^ mice demonstrated that deficiency of TCOF1 leads to apoptosis of neuroepithelial neural crest precursors, which decreases the population of neural crest cells migrating into the first and second pharyngeal arches leading to incomplete development of involved structures [4]. More recent work has extended these findings to show that the ubiquitination of TCOF1 (and analog NOLC1) promotes the association of RNA polymerase 1 with ribosome modification enzymes leading to altered translation of specific mRNAs and promotion of neural crest cell specification [58]. Together, these results suggest that strategies developed to decrease the ubiquitination of residual non-mutant TCOF1 could diminish severity of individuals with Treacher Collins syndrome caused by loss of function mutations in *Tcof1*.

Cleft palate is also associated with abnormal development of cranial neural crest cells and their derivatives. Palate development is a complex process that can be influenced by both environmental exposures [59] and inherent genetic abnormalities, well summarized in a review by Burg et al., 2016 [60]. Cleft palate exhibits high heritability [61,62]. In the United States, cleft palate in isolation occurs in one in 1700 births, while cleft palate with cleft lip occurs in 1 in 1600 birth [63]. Cleft palate can include the cleft of the hard and/or soft palate. Palate closure is essential for swallowing, speech, hearing, and breathing. The palatal shelves are composed of neural crest-derived mesenchyme surrounded by a layer of epithelium [64]. Signaling to and from cranial neural crest derived cells involving Wnt, FGF, and Hh (Hedgehog) signaling is essential for proper soft palate development [65]. *Pax9* is an essential transcription factor that is expressed in cranial neural crest-derived palate mesenchyme that is known to be essential for the prevention of cleft palate [66]. Repair of the soft palate can be compromised by lack of adequate muscle fibers for ideal surgical repair, leading to the potential for life long physical and psychosocial morbidity for affected individuals.

Pierre Robin sequence is a constellation of craniofacial anomalies that can occur in isolation, as part of an established syndrome or in association with other abnormalities that do not constitute an established syndrome [67]. Infants that are born with this disorder exhibit a small mandible (lower jaw) with posterior displacement of the tongue, cleft palate and upper airway obstruction due to the retracted tongue position [68]. Mouse model studies in which ERK2 (MAPK1) or BMP2 was conditionally deleted in cranial neural crest cells showed that the primary defect in this sequence of abnormalities is the deficient mandibular size, which leads to posterior tongue displacement and then to cleft palate [5,69]. ERK2 deficiency in cranial neural crest cells led to an early osteogenic differentiation defect, while BMP2 deficiency in cranial neural crest cells led to proliferation and differentiation defects, both specific to the mandible. The Pierre Robin sequence can therefore also be caused by diminished signaling in cranial neural crest cells leading to reduced proliferation and/or osteogenesis within the mandible. Notably, treatment for cleft palate, Treacher Collins and Pierre Robin sequence currently require surgical intervention that may not fully correct the involved craniofacial defects leading to medical and psychosocial morbidity for affected individuals [70,71].

Defects in cranial neural crest cells and their cell lineages also contribute to the pathogenesis of craniosynostosis (Table 1). Cranial sutures are the mesenchymal soft tissue that exist between growing cranial bones. Craniosynostosis is the pediatric condition in which cranial suture tissue is prematurely lost and cranial bone fusion occurs. This disorder occurs in approximately one per 2000 live births [72]. Premature cranial bone fusion due to loss of suture tissue causes high intracranial pressure as a result of limited growth at regions of fusion, and an abnormal craniofacial shape as a result of limited growth at regions of fusion combined with compensating overgrowth at regions of non-fusion (Figure 2). Craniosynostosis can also cause dental malocclusion, blindness, seizures, and death [72,73,74,75,76]. The severity of the craniofacial phenotype depends upon the timing of onset and on the number of sutures affected, with earlier onset and involvement of more cranial bones/sutures leading to more severe phenotypes [77]. Because the sole treatment is surgery, individuals with severe phenotypes suffer high morbidity. Some individuals require repetitive surgeries that increase the medical and financial burden [78,79,80]. Surgical approaches do not fully correct abnormal skull and facial shapes that contribute to psychosocial challenges [81].

Approximately 80% of individuals with craniosynostosis are non-syndromic, occurring with no additional anomalies or identified genetic abnormality, while approximately 20% of infants develop craniosynostosis as part of an established genetic syndrome. Non-syndromic craniosynostosis most commonly presents as fusion of the sagittal, metopic, or coronal sutures, while syndromic craniosynostosis involving single gene mutations most commonly involves the fusion of the coronal suture [82,83]. Many scientists study single gene mutation forms of syndromic craniosynostosis to gain insight into pathogenic mechanisms that may be relevant to all forms of craniosynostosis and because individuals with syndromic craniosynostosis tend to have more severe phenotypes. Studies utilizing genetic mouse models of syndromic craniosynostosis have shown that the pathogenesis can include pre-ossification development defects, such as insufficient differentiation of migrating cranial neural crest cells into mesenchymal cells, deficient neural crest cell renewal, and inappropriate mesoderm and/or neural crest cell localization to sites of bone and suture formation [10,84,85]. Craniosynostosis pathogenesis can also include post-ossification defects in cranial bone/suture boundary maintenance, premature cranial progenitor cell lineage commitment and suture osteogenesis, diminished proliferation, and/or apoptosis of suture stem or cranial bone progenitor cells [86,87,88,89,90,91,92,93,94]. Mechanisms differ, depending upon the genetic abnormality and involved suture. Despite these important advancements, unifying mechanisms and pharmacologic treatment options for craniosynostosis are not yet realized.

## 4. Cranial Neural Crest Cells in the Development of Cranial Bones and the Coronal Suture

Cranial neural crest cells are essential for the proper development of the craniofacial skeleton [26]. Cranial neural crest cells differ from other neural crest cells, in that they have the ability to differentiate into cartilage and bone. The first pharyngeal arch and domains just anterior to the first pharyngeal arch lack homeobox (*Hox*) gene expression and instead express *Msx* and *Dlx* homeobox domain containing transcription factors, enabling cartilaginous, and osteogenic potential [95]. When *Hox* genes are expressed in the first pharyngeal arch, craniofacial cartilage, and bone severely fail to develop [9].

Craniofacial bones and sutures are derived from the cranial neural crest and paraxial mesoderm [96] (Figure 3). Cranial neural crest cells from the mid-hindbrain region migrate to the anterior supraorbital arch region located directly above the developing eye between E8.0–E9.5 [97,98]. Paraxial mesoderm cells migrate with neural crest cells [25,97] from the mid-hindbrain towards the posterior supraorbital arch. Within the supraorbital arch region, neural crest and mesoderm cells both condense into respective mesenchymal progenitor cell populations. Neural crest cells transition to mesenchymal progenitor cells with the upregulation of *Sox9* and downregulation of *Sox10* transcription factors. This change in Sox transcription factors causes neural crest cells to lose their neural potential and gain the ability to eventually differentiate into craniofacial bone and cartilage [99,100,101]. The expression of Sox9 is also essential for neural crest cell proliferation leading to mesenchymal condensations [102,103].

Between E10.5 and E12.5, mesenchymal progenitor cells of neural crest and mesoderm lineages condense to respectively form frontal and parietal bone rudiments (primordial cranial bones) [96,97,104]. At this stage, progenitor cells express bone lineage markers *Msx1, Msx2, Runx2, Sp7, and En1* [105]. Expression of these transcription factors is required for establishing progenitor cells for the intramembranous ossification of cranial and facial bones. *Msx1* and *Msx2* are first expressed throughout the supraorbital arch mesenchyme at E10.5, but are later restricted to cranial bone progenitor cells at E12.5 [104,106,107]. The *Msx* transcription factors are required for the proliferation and differentiation of osteogenic cells within the developing cranial bone rudiments [106,107], in addition to the suppression of osteogenic differentiation in normally non-osteogenic cells near the growing rudiments [104]. Runt-related transcription factor 2 (*Runx2*) is a transcription factor that is necessary for early stages of osteogenic differentiation [108]. The premature expression of *Runx2* in cranial mesenchyme results in premature mineralization and prenatal fusion of cranial sutures in mice [109], whereas the loss of *Runx2* results in the complete lack of bone ossification [110]. *Sp7* acts downstream *Runx2* and it is required for osteoprogenitor lineage commitment at later stages of differentiation [111]. *Sp7* is required for intramembranous ossification of the developing cranial bones. *Twist1* encodes a basic helix-loop-helix transcription factor that regulates both FGF and BMP signaling [112,113,114]. It is required for mesenchyme condensation and the initiation of cranial bone rudiments [115,116,117]. At later stages of development, *Twist1* is a key regulator of suture cell proliferation and osteoblast differentiation [118]. *En1* is a homeobox domain containing transcription factor that is essential for proper neural crest/mesoderm boundary formation of the coronal suture, and also for preventing premature osteoblast lineage commitment of suture progenitor cells [9]. It is worth noting that *En1* is in the same pathway as *Twist1*, *Msx2,* and *Fgfr2* in craniofacial bone and suture development [9,10,85], such that this group of genes, when abnormally expressed, likely represents an essential developmental pathway to control coronal suture fusion.

Around E14.0–E14.5, frontal and parietal bones expand baso-apically and begin to mineralize. Proliferating cells from the frontal and parietal bone rudiments migrate to their respective growing osteogenic fronts [85,97,104]. Along the osteogenic fronts, cells differentiate into osteoblasts and lay down new bone, which causes the cranial bones to apically grow into the unossified suture mesenchyme. FGF signaling has an important regulatory role in this process. Proliferating cells along the osteogenic fronts express *Fgfr2* and differentiating cells basal to the osteogenic front express *Fgfr1* [119,120]. As stated above, the distinction between the proliferating and differentiating cells is also maintained by *En1, Twist1,* and *Msx2* as the cranial bones grow in this manner. The characteristic overlap of frontal and parietal bones separated by coronal suture mesenchyme is first observed around E17.5 [96,121] and the bones are mineralized at the lateral domains [121]. The coronal suture is first established between E9.5–E10.5 in mice as the caudal boundary of the frontonasal mesenchyme [9,96]. Fate mapping studies in mice have shown the coronal suture is of mesodermal origin [9,96,97]; however, it has been noted that the boundary between frontal and parietal bones is more complex than originally described as a pure mesodermal population of cells. While the coronal suture is embryonically derived from the mesoderm, lineage tracing studies have shown that the coronal suture exists as a mixed population of neural crest and mesoderm-derived cells in newborn mice [122]. The neural crest derived sagittal suture is first evident at E15.5 as the most caudal portion of the anterior neural crest domain becomes inserted at the midline between the paired parietal bones via anterior growth of the laterally positioned parietal bones into the neural crest domain [96]. As the parietal bones grow superiorly, the sagittal suture becomes more evident, remains derived from neural crest, and is adjacent to mesoderm derived parietal bones and neural crest derived meninges [96].

With continued growth and ossification, the cranial bones form close proximal relationships. At this point, bone growth continues to occur via osteogenesis within the osteogenic fronts while still maintaining suture patency. Cranial suture patency is commonly maintained throughout growth and is thought to allow for distortion of the skull during birth, dampening of mechanical forces (to decrease injury), and compensation for the expansive forces of the growing brain. It is important to note that the cells within the established coronal suture are not all the same. Midline suture cells behave as suture stem cells with low renewal rates, while the lateral suture contains osteoprogenitors that exhibit high rates of proliferation with differentiation into osteoblasts in the osteogenic front [105]. Therefore, defects in either suture stem cells or suture osteoprogenitors can contribute to craniosynostosis late in embryonic development or after birth while the skull is still growing.

## 5. Mechanisms Underlying Coronal Craniosynostosis

Craniofacial anomalies, and among them craniosynostosis, must be understood both in terms of a genetic mutation’s molecular effect on an individual cell and the collective interactions between cell populations of different embryonic lineages. The coronal suture is a unique structure in terms of its formation, development, and location surrounded by cranial bones of different embryonic origin, as described above. Much of the knowledge we have regarding coronal suture development comes from studying the progression of suture fusion in mouse models of human syndromic craniosynostosis. The classic human craniosynostosis syndromes and associated genetic mutations involving coronal suture fusion include Pfeiffer and Jackson–Weiss syndromes (*FGFR1*) [123,124,125]; Pfeiffer, Apert, Crouzon, Jackson-Weiss, and Beare-Stevenson syndromes (*FGFR2*) [126,127,128,129,130,131]; Muenke syndrome (*FGFR3*) [132,133]; Saethre-Chotzen syndrome (*TWIST1*) [134,135]; craniofrontonasal syndrome (*EFNB1*) [136,137]; and, Boston-type craniosynostosis syndrome (*MSX2)* [138,139,140]. (Table 1). FGF signaling and associated transcription factors *Twist1* and *Msx2* are particularly important for the development of the coronal suture and surrounding cranial bone, so it is not surprising that mutations in the corresponding genes lead to premature coronal suture fusion.

An often-posed question is whether such mutations have an embryonic origin-specific effect on the cranial bones that either directly or indirectly induce coronal suture fusion. Lineage-restricted expression of Apert mutant *Fgfr2^S252W^* in mesoderm-derived tissue (including the parietal bone and coronal suture) was necessary and sufficient to cause coronal synostosis, whereas restricted expression in cranial neural crest-derived tissue was not [93]. When *Twist1* function is lost in the mesoderm lineage, the posterior cranial vault and cranial base bones fail to develop, but the neural crest-derived cranial vault and base bones are also reduced in size [141]. While the mesoderm-derived tissue of the skull cannot be ignored when discussing coronal craniosynostosis, we will focus our subsequent discussion as to how abnormal development of neural crest-derived cells contributes to the pathogenesis of coronal suture fusion.

**Table 1 jdb-08-00018-t001:** Neural Crest Mechanisms Underlying Coronal Craniosynostosis.

Human Syndrome	Associated Mouse Model Genetic Mutation	Human Craniofacial Phenotype	Proposed Mechanism(s) of Anomaly
Craniofrontonasal Syndrome (OMIM #304110)	*Efnb1^−/−^*	anterior-posteriorly shortened skull, facial dysmorphologies, coronal suture fusion *	Neural crest-specific disruption of *Efnb1* disrupts lineage-based boundary formation of coronal suture [137,142].
Apert Syndrome (OMIM #101200)	*Fgfr2* ^S252W/+^	Coronal, sagittal, lambdoid suture fusion; proptosis, hypertelorism, midface hypoplasia	Enhanced osteogenic differentiation along osteogenic front of parietal bone enhanced by neural crest-derived frontal bone [91,93,126].
Crouzon Syndrome (OMIM #123500)	*Fgfr2* ^C342Y/+^	Coronal suture fusion, proptosis, hypertelorism, midface hypoplasia	Embryonic dysregulation of Sox9 expression causing mesenchymal condensation defects, symptoms of neural tube defects, plus decreased craniofacial osteogenesis and increased chondrogenesis; postnatal enhanced osteogenic differentiation within osteogenic fronts; [84,87,130].
Muenke Syndrome (OMIM #602849)	*Fgfr3^P250R/+^*	Coronal suture fusion; pansynostosis; hearing loss; midface hypoplasia	Hearing loss due to embryonic fate switch of neural crest derived cochlear Deiters’ cells to pillar cells [143,144].
Bent Bone Dysplasia (OMIM #614592)	*FGFR2^C1172T^* ^Φ^	Coronal suture fusion; midface hypoplasia; prenatal teeth; low set ears; micrognathia; diminished bone mineralization; bent long bones	Mutations promote ribosomal transcription within the nucleus leading to enhanced osteoprogenitor cell proliferation with diminished differentiation [145,146].
Saethre-Chotzen Syndrome (OMIM #101400)	*Twist1* ^+/−^	Coronal suture fusion, low hairline, hypertelorism, ptosis, broad nasal bridge, digit fusions	Disruption of lineage-based boundary formation of coronal suture and cell lineage mixing. Enhanced osteogenic potential of parietal vs. frontal bones [10,90,135,147].
TCF12 (OMIM # 600480)	*Tcf12^+/−^/Twist^+/−^*	Described as a milder form of Saethre-Chotzen syndrome. Coronal suture fusion, facial dysmorphologies, minor limb abnormalites	TCF12 is dimerization partner for TWIST1. Double mutant mice show accelerated parietal and/or frontal bone growth plus diminished pool of osteoprogenitors in coronal suture [147,148].
Non-Syndromic Coronal Synostosis	*EphA4*^−/−^ and *Twist1*^+/^/*EphA4*^+/−^	Coronal suture fusion	Disruption of boundary formation and neural crest/mesoderm cell lineage mixing due to lack of *Twist1* and its effector *EphA4* [85].
Infantile Hypophosphatasia (OMIM #241500)	*Alpl* ^−/−^	Coronal or sagittal suture fusion ^#^, hypomineralization, midface hypoplasia.	Hypomineralization and cell proliferation defects more severe in cells of neural crest derived craniofacial bones; enhanced FGFR2 signaling in osteoprogenitors; [149,150].

* The *Efnb1*^−/−^ mouse craniofacial phenotype does not correspond to the human craniofacial phenotype. The *Efnb1*^−/−^ mouse has an anterior-posteriorly shortened skull but does not have craniosynostosis, as is seen in individuals with Craniofrontonasal Syndrome. # The *Alpl*^−/−^ mouse model of hypophosphatasia develops coronal but not sagittal suture fusion, while fusion of coronal or other cranial sutures may be evident in infants with this metabolic disorder. Φ The Bent Bone Dysplasia mutation in FGFR2 is a human mutation that has been studied in vitro (to our knowledge, no mouse model has yet been developed).

### 5.1. The Impact of Embryonic Origin on Cranial Bone and Coronal Suture Development

Inherent differences between the neural crest-derived frontal bone and paraxial mesoderm-derived parietal bone have been extensively investigated. First, frontal bone osteoblasts are more proliferative and more osteogenic than osteoblasts from the parietal bone [122,151,152]. Frontal bone osteoblasts have higher osteogenic gene expression (*Runx2, Alpl, Bglap*), which suggests a greater pool of osteoprogenitors resides in the frontal bone [121,151]. Parietal bone osteoblasts co-cultured with frontal bone osteoblasts become more proliferative, more osteogenic, and able to contribute to bone nodule formation [122,151]. Second, neural crest-derived craniofacial bones represent domains of activated FGF signaling [151,153], which could predispose these tissues to being more susceptible to mutations in FGFRs. This could explain why FGFR-related craniosynostosis syndromes often include frontal and facial bone defects. Third, neural crest-derived osteoblasts have greater regeneration potential than paraxial mesoderm-derived cells due enhanced canonical Wnt signaling [153]. Taken together, these results indicate that neural crest-derived cells may be more proliferative and osteogenic in nature within activated Wnt and FGF signaling domains, and they can influence mesoderm-derived osteoblasts. This pro-osteogenic influence could potentially cause unwanted differentiation of the mesoderm-derived coronal suture and result in suture fusion.

Movement and/or loss of the coronal suture can occur if factors further enhance or diminish frontal vs. parietal bone osteogenesis, leading to imbalanced growth between these bones. Recent findings show that haploinsufficiency of *Twist1* in mouse neural crest (*Wnt1-Cre*; *Twist1*^fl/+^ mice) leads to posterior positioning of the coronal suture due to increased frontal over parietal bone growth, while *Twist1* reduction in mesoderm (*Mesp1-Cre*;*Twist1*^fl/+^ mice) leads to anterior positioning of the coronal suture due to increased parietal over frontal bone growth [147]. Haploinsufficency of *Twist1* in both neural crest and mesoderm lineages led to a loss of the coronal suture. Similar findings were established in zebrafish [154]. The fact that the coronal suture in zebrafish and mice develops from different embryonic origins suggests that diminished or excessive osteogenic potential of frontal or parietal bones can lead to coronal craniosynostosis independent of coronal suture embryonic origin. In support of the idea that changes in parietal and frontal bone growth can cause loss of the coronal suture, in ciliopathic mutant mice the coronal suture is absent because the frontal bone, but not parietal bone, develops and grows to encase the forebrain [35]. Such a differential cranial bone growth phenomenon would likely apply to the coronal, but not sagittal suture, as the sagittal suture lies between paired parietal bones of same embryonic origin and osteogenic potential.

### 5.2. Defects in Neural Crest-Derived Progenitor Cell Proliferation, Differentiation, and Survival

It is not surprising that FGFR-associated craniosynostosis syndromes have defects in neural crest-derived skeletal tissues when considering the fact that neural crest-derived cranial bone progenitor cells are more proliferative, more osteogenic and exist as domains of greater FGF signaling potential. Crouzon syndrome is the most common of the FGFR2 craniosynostosis syndromes [155]. It occurs most commonly due to “gain-of-function” mutations in the mesenchymal splice variant of *FGFR2c*, in which a cysteine residue in the third immunoglobulin-like domain is eliminated (C342Y or C278F), resulting in ligand-independent intramolecular receptor dimerization [156,157,158].

Crouzon syndrome is characterized by the distinct “Crouzonoid” appearance—bicoronal suture fusion with occasional pansynostosis, severe midface hypoplasia, hypertelorism, and severe ocular proptosis [159]. Unlike Pfeiffer and Apert syndromes, there are no limb or digit abnormalities in individuals with Crouzon syndrome, suggesting that the mutation has greater specificity of effects on neural crest-derived craniofacial tissues. Indeed, lineage-restricted over-expression of *Fgfr2c* in neural crest-derived tissues causes midface hypoplasia and cleft palate, whereas lineage-restricted over-expression in mesoderm-derived tissues yields no craniofacial phenotype [86]. The *Fgfr2*^C342Y/+^ mouse model phenocopies the craniofacial abnormalities seen in individuals with Crouzon syndrome, including the hallmark features of coronal and facial suture fusion, midface hypoplasia, hypertelorism and ocular proptosis [87,130]. Prior work with this mouse model has shown the *Fgfr2*^C342Y^ mutation promotes premature osteogenic lineage commitment, inhibits mineralization, and induces apoptosis in neural crest-derived primary cells and tissues prior to the onset of coronal suture fusion [87,130,160]. More recent work demonstrates that Sox9 is dysregulated during embryonic development of the Crouzon mouse and that this leads to defects in mesenchyme condensations plus decreased osteogenesis and increased chondrogenesis in the craniofacial skeleton [84]. Why and how the Crouzon phenotype is commonly restricted to bones of the neural crest lineage, and whether or not these defects are driving coronal suture fusion, remain unanswered questions.

Similar to Crouzon syndrome, individuals with Apert syndrome have bicoronal suture fusion and midface hypoplasia, albeit more severe than in Crouzon syndrome [126]. Apert syndrome has additional distinguishing features, including syndactyly of the hands and feet, brain abnormalities, and sometimes cleft palate and hearing loss [161,162]. Apert syndrome is caused by a missense mutation in FGFR2 (S252W or P253R) in the linker region between the second and third immunoglobulin-like domains [126,131,163]. These mutations increase binding affinity and decrease specificity of FGFR2c for FGF ligands [164,165]. Apert cranial progenitor cells are more osteogenic and prematurely apoptotic prior to the onset of coronal suture fusion [91,160,166,167,168]. As noted above, lineage-restricted expression of Apert *Fgfr2*^S252W^ in the mesoderm, not the neural crest, results in coronal suture fusion [93]. Understanding that, in normal development, the neural crest-derived frontal bone has superior osteogenic potential that can promote the osteogenic differentiation of mesoderm-derived parietal bone cells [122,151], it seems logical to hypothesize that a more osteogenic frontal bone could exacerbate the osteogenesis of *Fgfr2*^S252W/+^ mutant mesoderm derived tissue to cause the Apert craniofacial phenotype, as is seen in the mesoderm specific mutant mouse. This is an example of how the osteogenic potential of one embryonic lineage can impact the other to contribute to the progression of craniosynostosis.

### 5.3. Boundary Defects between Developing Cranial Bones

The fact that the coronal suture exists as a mixed population of neural crest and mesoderm-derived cells at some stages of development, and it sits between bones of neural crest and mesoderm origin [96,97,122,169], is important to recognize for our understanding of the complex mechanisms that control suture development and patency. Cell lineage mixing within the coronal suture mesenchyme is another manner by which neural crest and mesoderm-derived tissues (frontal and parietal bones, respectively) can contribute to premature coronal suture fusion. In normal development, the defined neural crest/mesoderm boundary of the coronal suture must be tightly regulated through FGF signaling (inclusive of *Twist1, En1,* and *Msx2* transcription factors) in order to prevent cell lineage mixing. The mixing of neural crest with mesoderm derived progenitors is a potential mechanism of cranial suture fusion. In this scenario, neural crest derived cells invade the suture mesenchyme with osteogenic cells and increase the likelihood for differentiation into bone [85]. Mutations in FGFR2 have not been shown to cause lineage mixing within the suture mesenchyme; however, both the *En1*^−/−^ mouse model and the *Twist1*^+/−^ mouse model of Saethre-Chotzen syndrome demonstrate extensive lineage mixing leading to coronal suture fusion [9,10]. The *Twist1* transcription factor is required for establishing the coronal suture boundary and regulating osteoprogenitor cell proliferation and differentiation through inhibiting *Fgfr2* expression and terminal osteoblast differentiation [112,118,170]. *En1* regulates the neural crest/mesoderm boundary formation of the coronal suture through *Twist1* and *Msx2*. As a downstream effector of *Twist1, Msx2* works cooperatively to control proliferation and differentiation of neural crest-derived osteogenic cells of the frontal bone rudiment [106,107]. The loss-of-function of *En1* results in an embryonic caudal shift of this boundary, mixing of neural crest-derived cells in the coronal suture and parietal bone, as well as expanded *Msx2* and diminished *Twist1* expression around the prospective suture [9]. The loss of *Twist1* in the suture mesenchyme also causes cell lineage mixing and coronal suture fusion in the *Twist1*^+/−^ mouse model of Saethre-Chotzen syndrome [10]. Saethre-Chotzen syndrome is characterized by unicoronal or bicoronal fusion, facial asymmetry, hypertelorism, and maxillary hypoplasia [134,135]. In addition to cell mixing in the suture mesenchyme, loss of *Twist1* widens *Msx2 expression* distribution and osteogenic differentiation [10,90,114]. Loss of *Twist1* also reduces the levels of ephrin receptor *EphA4* and ephrin ligands ephrin-A2 (*Efna2)* and ephrin-A4 (*Efna4*), all of which are components of the eph/ephrin signaling pathway that is crucial for tissue boundary formation in vertebrates [10,85,171]. Reduced dosage of *Msx2* in *Twist1^+/−^* mice can rescue the coronal suture boundary defect to a wild type pattern [10]. In humans, heterozygous mutations in *EFNA4,* loss-of-function of *EFNB1* in craniofrontonasal syndrome, and the amino acid substitution P148H in *MSX2* of Boston-type craniosynostosis syndrome all result in coronal suture fusion in humans [10,136,137,139,140], which further indicates the necessity of coronal suture boundary maintenance for healthy skull development.

### 5.4. Premature Loss of Suture Stem Cells

Aberrant cranial progenitor cell proliferation, premature or ectopic osteoblast differentiation and/or increased apoptosis are the most established potential mechanisms of coronal craniosynostosis. It is not currently known whether early embryonic loss of mesoderm or cranial neural crest cells, loss of cranial suture stem cells within the midline of the suture [105], and/or loss of cranial bone osteoprogenitors within the osteogenic fronts of growing cranial bones [120] can all contribute to the phenotype. While not fully established, there is strong evidence supporting the idea that premature loss of suture stem cells is a mechanism behind craniosynostosis. First, lineage tracing techniques have identified *Gli1* and Axin2-expressing mesenchymal cell populations in the cranial sutures that are capable of stem cell-like behaviors, including clonal expansion, multipotent differentiation, and injury repair [172,173]. Axin2-expressing cells are restricted to the midline of the suture and overlap with *Gli1*+ cells, which make up most of the suture mesenchyme. The loss of *Gli1*+ cells in the coronal suture likely changes the landscape of the neural crest/mesoderm boundary as this was observed in all sutures of *Twist1*^+/−^ mice [172]. As of writing this review article, a true suture stem cell population has yet to be isolated and characterized. Identifying a putative suture stem cell has great potential for tissue engineering and translational medicine applications to improve craniosynostosis patient surgical outcomes.

### 5.5. Epigenetic Influences on Craniosynostosis

Epigenetic changes, such as open vs. closed chromatin states, absence or presence of DNA binding proteins, topological association of chromatin domains, and genetic networks control transcriptional programs, including those that are specific to cell fate decisions and craniofacial development. Histones and histone modifying enzymes control chromatin compaction and the availability of DNA to transcriptional repressors and enhancers, thereby influencing the transcriptional programs needed for cell replication and development [174,175]. One example of a histone modifying enzyme that is relevant for craniofacial development is *Ezh2* (enhancer of zeste homolog 2). EZH2 is a histone 3 (H3) modifying enzyme that acts as a transcriptional repressor by methylating histone 3 on lysine 27 (H3K27 methylation) [176,177]. Of relevance to craniosynostosis, studies of EZH2 have demonstrated that it plays an essential role in neural crest cells and craniofacial development. Genetic ablation of *Ezh2* in neural crest (using *Wnt1-Cre*) prevents craniofacial bone and cartilage formation in mice due to the de-repression of Hox genes that would not normally be expressed in cranial neural crest cells [101]. The conditional knockout of *Ezh2* in mesenchymal precursors (using *Prrx1-Cre*) causes coronal craniosynostosis [178]. Interestingly, the coronal craniosynostosis phenotype is not apparent in mice, in which *Ezh2* is ablated in osteoblasts (using *Sp7-Cre*) [179]), nor is it observed upon the loss of *Ezh2* in cartilaginous tissues (using *Col2-Cre*) [180]. *Ezh2* expression also correlates with osteoblast differentiation stages (expression decreases with differentiation), and controls the switch between osteoprogenitor proliferation vs. differentiation [179,181]. Therefore, EZH2 may play both an early role in embryonic neural crest cell development and a later role in craniofacial osteoprogenitors to influence craniofacial development and coronal suture fusion. EZH2 can also influence topologically associating domains of chromatin, areas which tend to be marked by H3K27 methylation/demethylation [182,183,184]. It should be noted that EZH2 mutations in humans cause Weaver syndrome, a phenotype that includes distinct facial dysmorphologies, but not craniosynostosis [185,186]. To take into account potential genetic and epigenetic differences between mice and humans, and to investigate mechanisms that may be causal for coronal craniosynostosis, one can utilize online databases. Examples include a database generated by a comprehensive epigenetic analysis of human embryonic craniofacial tissues [187] and the Ontology of Craniofacial Development and Malformation that is comprised of a genetic and epigenetic database that includes both mouse and human craniofacial structures and developmental time points [188]. Spatiotemporal transcriptome analysis of wild type and mutant mice craniofacial tissues provides important information on cranial bone and suture cell subpopulations and biological processes, including angiogenesis and ribogenesis, which are central to suturogenesis [189]. Clearly, many other factors in addition to EZH2 are likely to influence craniofacial development and coronal suture fusion via epigenetic mechanisms. In addition, it should be noted that craniosynostosis genes function within a network, such that a single gene mutation can alter expression and/or function or many other genes within that network, ultimately leading to the phenotype of craniosynostosis [190]. There are currently 3,164 genes in Online Mendelian Inheritance in Man^®^ listed under the search term “craniosynostosis causing”, indicating that that we are perhaps at the tip of the iceberg in terms of understanding the craniosynostosis genetic network [191].

## 6. Development of Strategies for Treatment and Future Outlook

Individuals with craniofacial anomalies require surgical intervention. Individuals with severe phenotypes can experience high morbidity due to the need for multiple corrective surgeries throughout childhood and adolescence [78,79,80]. Craniofacial surgeons also encounter challenges with not having enough tissue to work with in these reconstructive surgeries. In this century, advances in the field of tissue engineering have shown promise in developing polymer scaffold-based systems for use in craniofacial surgery. Scaffolds can be highly tuned for specific applications that are based on material choice, fabrication method, and surface functionalization [192,193]. Pertinent to craniofacial surgery, synthetic scaffolds can be designed to maintain an undifferentiated stem cell niche for a cranial suture or promote osteogenic differentiation to heal bone defects [194,195,196,197,198]. In the future, surgeons should be able to utilize scaffolds fabricated to control stemness and promote craniofacial growth for improved surgical outcomes, particularly when coupled with appropriate surgical techniques. For example, distraction osteogenesis can induce the transformation of mouse skeletal stem cells into neural crest stem cells via focal adhesion kinase (FAK) signaling to regenerate bone tissue [199]. While distraction osteogenesis as a surgical technique has limitations for craniofacial anomaly phenotype correction, the idea that mechanical forces and FAK signaling can be utilized in order to recreate a primitive neural crest cell phenotype in vivo lends great promise to the development of treatments, including tissue regeneration strategies for craniofacial anomalies of cranial neural crest origin.

Surgical approaches cannot always fully correct craniofacial anomaly phenotypes. Therefore, it is important that investigators continue to focus on understanding molecular mechanisms behind pathogenesis with a focus on the development of novel non-surgical strategies for prevention or treatment. Our research group is investigating how tissue non-specific alkaline phosphatase (TNAP/*Alpl*) in Crouzon *Fgfr2*^C342Y/+^ mice can diminish the severity of coronal suture fusion and craniofacial defects. Similar to Crouzon *Fgfr2*^C342Y/+^ mice, TNAP-deficient mice exhibit coronal suture fusion, a dome shaped skull, hypertelorism, and severe midface hypoplasia [87,130,149,200]. Numerous previous studies demonstrated that FGF signaling regulates TNAP expression in a cell type and differentiation stage dependent manner [87,201,202]. More recent studies indicate that TNAP deficiency increases FGFR2 expression and diminishes cranial progenitor cell cycle progression and proliferation [150]. Of relevance to patient care and reduction in morbidity, we recently showed that lentiviral delivery of recombinant TNAP to *Fgfr2*^C342Y/+^ mice shortly after birth diminished the severity of coronal suture fusion and incidence of class III malocclusion [203] (Figure 4a,b). While not a complete rescue, the data do suggest that TNAP may be efficacious for diminishing the severity of FGFR2-associated coronal craniosynostosis. Support for the hypothesis that one function of TNAP is to promote progenitor cell development is evidenced by the fact that TNAP is expressed in pluripotent primordial germ cells during early embryonic development [204,205,206], in cranial neural progenitor cell populations [207], and in cranial bone rudiments several days prior to the onset of matrix mineralization [208,209]. Notably, neural crest derived bones are more severely hypomineralized in TNAP deficient mice (Figure 4c,c’). We are currently generating neural crest specific TNAP knockout mice in order to determine if and how TNAP deficiency in cranial neural crest cells and/or their derivatives leads to coronal craniosynostosis. This is but one example of how the investigation of molecular mechanisms behind specific craniofacial anomalies is essential for the development of disease-modifying therapies.

As craniofacial biologists, we can gain inspiration from the cancer literature when investigating novel prevention and treatment strategies for FGF/FGFR associated craniofacial anomalies due to the prevalence of FGFR “gain-of-function” mutations in both craniosynostosis syndromes and various cancers [210,211,212]. Small molecule inhibitors are often used in cancer therapy [213]. Small molecule inhibitors of FGFR and downstream ERK1,2 signaling have shown promise in preventing coronal suture fusion in Apert *Fgfr2*^S252W/+^ mice and Crouzon *Fgfr2*^C342Y/+^ cranial vault organ culture [214,215,216]. Recently, a less toxic small molecule FGFR tyrosine kinase inhibitor (ARQ 087) was developed and it is in clinical trials for its potential use in the treatment of human cancers [217,218]. Cranial vault organ culture studies of ARQ 087 show that it is efficacious in preventing coronal suture fusion due to FGF2-triggered excessive osteogenic differentiation [219]. In addition to FGFR inhibitors, therapeutic inhibitors that target the intracellular protein quality control system (including heat shock/chaperone proteins and ubiquitin ligases) are also considered to be a viable option for some cancer treatments [220,221,222]. Various heat shock proteins are known to protect oncogenic and mutant proteins from misfolding and degradation, thereby promoting the evasion of intracellular protein quality checkpoints and cancer progression [223,224]. Several studies have investigated this fundamental biologic concept of protein quality control to explain the abnormal cell behavior and development associated with genetic mutations causal for craniofacial anomalies, such as syndromic craniosynostosis. Previous studies showed that Apert FGFR2(S252W), Crouzon FGFR2(C278F), and Boston-type MSX2(P148H) mutant proteins exhibit increased ubiquitination and degradation [167,225,226,227,228]. Future investigation should consider whether differences in intracellular protein quality control exhibit embryonic lineage specificity and how inhibiting or promoting the activity of protein quality control checkpoints influences cell behavior and craniofacial development.

Other recent laboratory advancements include more widespread use of sequencing technologies to generate detailed epigenomic and transcriptional atlases of tissues and cell populations. Chromatin immunoprecipitation coupled with DNA sequencing (ChIP-seq) and Assay for Transposase Accessible Chromatin sequencing (ATAC-seq) can be used in order to identify coding and non-coding regulatory elements in normal and pathological craniofacial development [187,229]. Single-cell RNA sequencing (scRNA-seq), when combined with in vivo lineage tracing, can provide detailed information on the spatiotemporal transcriptional landscape of individual cell populations during development. Recently, the use of these techniques in early mouse embryos revealed that mesenchymal fate decisions of cranial neural crest cells are already determined upon delamination from the neural tube [230]. Next generation sequencing technologies, including whole exome and whole genome sequencing on DNA from patients without identified mutations, can reveal new genes that are associated with the development of craniosynostosis [231]. Finally, improvements in in vitro model systems enable advances in fundamental knowledge of mechanisms controlling neural crest cell behavior and migration using live cell imaging [232] and differentiation using cranial neural crest cell lines [233]. The use of these advanced sequencing technologies with animal and cell models will aid in answering remaining questions in the field of craniofacial development, including but not limited to investigating phenotype variation among individuals with craniofacial anomalies caused by the same genetic mutation, determining how early in development a genetic mutation affects cell lineages and/or populations, further establishing how neural crest and paraxial mesoderm lineages influence each other during craniofacial development, defining a true cranial suture stem cell population and establishing methods for cranial neural crest and stem cell regeneration in vivo.

## 7. Conclusions

Craniofacial development is a complex, highly regulated process that requires the coordinated interaction of signaling between cells and tissues of two distinct embryonic origins. Advances in mouse genetic technologies for the generation of relevant mouse models and cell lineage tracing technologies have greatly increased our understanding of both healthy and pathological craniofacial development. The mouse model allows for spatial and temporal analysis of how cells and tissues of neural crest and paraxial mesoderm origin interact during craniofacial development. In vitro systems fail to model these complex relationships; however, they can serve as a useful tool for investigating signaling and molecular properties of proliferation and differentiation that may contribute to pathological development. In this review, we have discussed how cranial neural crest cells are essential for craniofacial development and how defects in cranial neural crest cells cause craniofacial anomalies. We have described the coronal suture as the biological boundary between the neural crest-derived frontal bone and paraxial mesoderm-derived parietal bone, how genetic mutations often lead to coronal suture fusion (craniosynostosis), and potential cellular mechanisms that are responsible for this. Surgery is the only available treatment for individuals with craniofacial anomalies. Novel, non-invasive treatment strategies may only be realized through continued advances in the field of craniofacial development. We can one day improve the quality of life and surgical outcomes for individuals with craniofacial anomalies by increasing our understanding of the basic cell and developmental biology behind pathological craniofacial development.

## Figures and Tables

**Figure 1 jdb-08-00018-f001:**
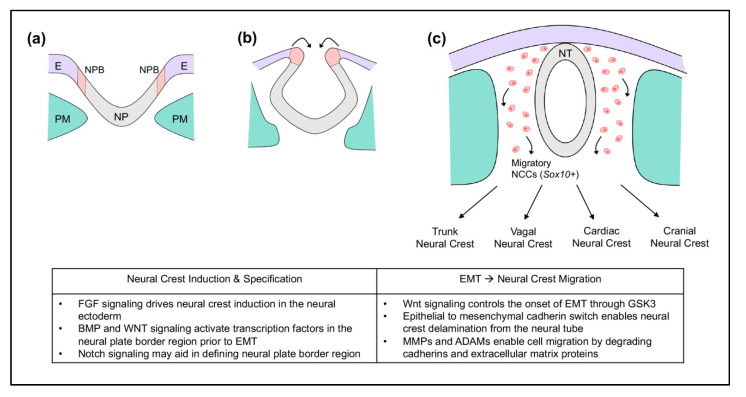
Early neural crest cell development in mouse embryos. (**a**) Gastrulation: neural crest cells are induced at the neural plate border (NPB) between the neural plate (NP) and non-neural ectoderm (E). Paraxial mesoderm (PM) is underlying. (**b**) Neurulation: the neural crest becomes specified. The neural plate begins to fold to later form the neural tube (NT). (**c**) Prior to neural tube closure in mouse, epithelial-mesenchymal transition (EMT) triggers neural crest cells to delaminate from the neural tube and migrate throughout the embryo. Migratory neural crest cells express *Sox10*.

**Figure 2 jdb-08-00018-f002:**
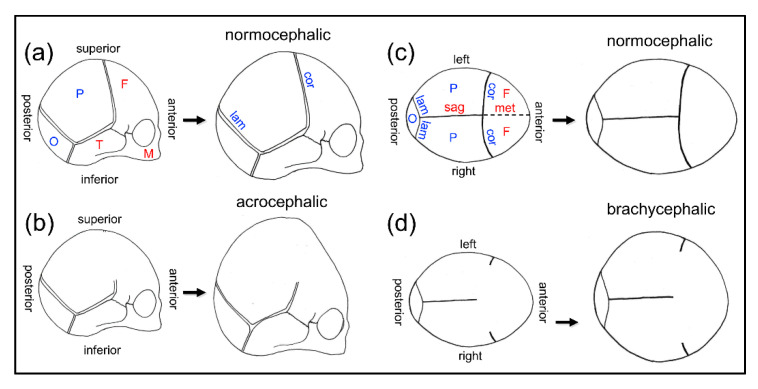
Coronal craniosynostosis influences skull growth and shape in humans. During late embryonic and early postnatal development, the skull increases in size via osteogenesis within the osteogenic fronts of each cranial bone. With no fusion of cranial sutures, the skull increases in size and maintains a normocephalic shape. Coronal, sagittal and lambdoid sutures remain patent, while the metopic suture fuses within months after birth (**a**,**c**). Upon coronal suture fusion, the anterior aspect of the parietal bone(s) fuses with the posterior aspect of the frontal bone(s) such that growth cannot occur in this region. This leads to an acrocephalic (taller relative to anterior-posterior length) (**b**) and brachycephalic (wider relative to anterior-posterior length) skull shape (**d**). Limited growth along the coronal suture also leads to compensating vertical and transverse overgrowth along other non-fused sutures, such as the sagittal and lambdoid sutures. Limited skull growth due to craniosynostosis causes high intracranial pressure which must be surgically relieved. sag = sagittal suture, cor = coronal suture, lam = lambdoid suture, met = metopic suture, O = occipital bone, P = parietal bone, F = frontal bone, T = temporal bone, M = maxillary bone. Bone and suture labels in blue are derived from paraxial mesoderm. Bone and suture labels in red are derived from cranial neural crest. Note: all facial bones and sutures are derived from neural crest.

**Figure 3 jdb-08-00018-f003:**
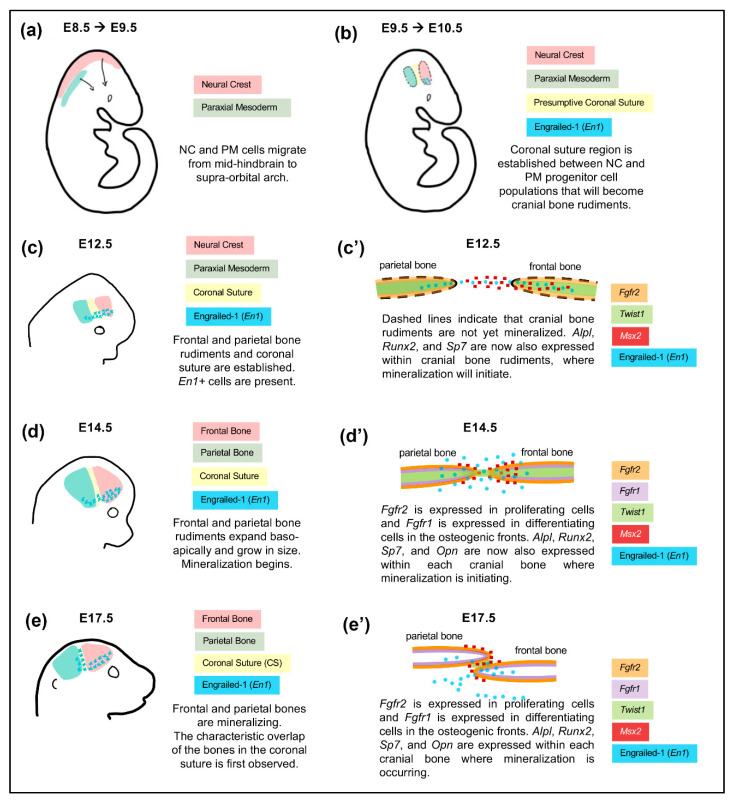
Embryonic Development of the Coronal Suture with Frontal and Parietal Bones. The coronal suture exists as a physical boundary separating the neural crest-derived frontal bone and paraxial mesoderm-derived parietal bone throughout embryonic development. (**a**–**e**) Schematics depict lateral view of whole skull. (**c’**–**e’**) Schematics depict sagittal sections of the skull lateral to the midline to describe the spatial relationship of the developing frontal and parietal bones. The expression patterns of genes essential for development are depicted in various colors at indicated time points.

**Figure 4 jdb-08-00018-f004:**
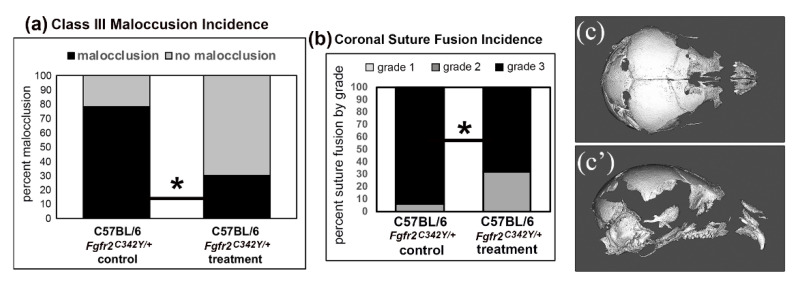
Postnatal Delivery of Tissue Nonspecific Alkaline Phosphatase (TNAP) Enzyme Diminishes Abnormal Craniofacial Phenotype of Crouzon *Fgfr2*^C342Y/+^ Mice. (**a**) Percentage of vehicle (control) and TNAP treated *Fgfr2*^C342Y/+^ mice with a class III malocclusion are shown. Results show a significantly decreased incidence of malocclusion in C57BL/6 *Fgfr2*^C342Y/+^ mice. (**b**) Percentage of vehicle (control) and TNAP treated *Fgfr2*^C342Y/+^ mice with coronal suture fusion are shown. Fusion was scored as: (1) diminished suture width with no fusion, (2) diminished suture width with point fusions across the suture, and (3) obliteration of the suture. Results show a significantly decreased incidence of suture obliteration in C57BL/6 *Fgfr2*^C342Y/+^ mice. * *p* < 0.03 between treatment groups. (**c**,**c’**) Axial and lateral isosurface images of *Alpl*^−/−^ (TNAP knockout) mice exhibit craniofacial bone hypomineralization that is more severe in bone of cranial neural crest than paraxial mesoderm origin. Note that numerous bones of neural crest origin are so hypomineralized that they do not appear on a micro CT image generated using a bone threshold. Adapted from [203].

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
