# Peer review of "Cranial Neural Crest Cells and Their Role in the Pathogenesis of Craniofacial Anomalies and Coronal Craniosynostosis"

_jdb, 2020, doi:10.3390/jdb8030018_

Round 1
Reviewer 1 Report
“The cranial neural crest cells and their role in the pathogensis of cranial anomalies” is an important topic which discuss about the role of cranial neural crest cells in the development of the facial birth defects. The neural crest cells are important in facial morphogenesis, specifically in craniofacial skeleton development. Here the authors have discussed the genetic syndromes associated with the cranial neural crest cell malfunction. The role of neural crest cells in the development of the sutures are discussed in the greater details. However there are several major issues which need to be addressed before this being considered for the publication.
This review does not particularly discuss the current knowledge of molecular and genetic studies and findings of different organisms such as avian and fish models in neural crest research.
My specific comments about this review is as follows.
Subtopic “Neural crest cells”
The origin of the neural crest cells are different in different organisms. Please be specific and include references about different mechanisms of origin of crest cells in different organisms.
Authors mentioned BMP, WNT and FGF as key cell signalling pathways of neural crest cell migration. This is bit misleading. Other than above Hh, Notch and TGF pathways signalling are important in regulation of neural crest cell fate. It is important to mention these as well.
Subtopic “Cranial neural crest cells in the development of cranial bones and coronal suture”
Line # 227-229 “coronal suture exists as an intramembranous bone growth site and a physical boundary between neural crest-derived frontal bone and paraxial mesoderm-derived parietal bone”. This information is repetitive. Check the manuscript and edit accordingly.
Subtopic “Molecular mechanisms underlying Coronal Craniosynostosis”
This section needs to be revised according to the title. What authors mean by molecular mechanisms are unclear. Here they discuss only genetic syndromes and cell signalling pathways. But needs to be include the epigenetic factors and the downstream signalling mechanisms which are critical in determining the neural crest cell fate.
The discussion about the lineage tracing studies are fine. But you can include other genetic studies such as knock down and knock in models, genome wide associated studies and studies done on other animal models.
Figure revisions
Figure 1. This needs to be labeled properly as it it difficult to understand a person who does not know the anatomy of the human skull. Indicate the directions and explain the view. ex: Anterior –Posterior, name the bones.
Figure 2.Enlarge the diagram. Label the sequence (a,b,c or 1, 11, 11)
Figure 3. The (a) and (b) are in wrong sequence and not match with the legend. Please correct the figure or the legend.
Author Response
We would like to thank this reviewer for their time and effort to do a thorough review of this manuscript. In response, many changes were incorporated. We hope that this reviewer agrees that the manuscript is improved as a result.
Reviewer 1
The cranial neural crest cells and their role in the pathogenesis of cranial anomalies” is an important topic which discuss about the role of cranial neural crest cells in the development of the facial birth defects. The neural crest cells are important in facial morphogenesis, specifically in craniofacial skeleton development. Here the authors have discussed the genetic syndromes associated with the cranial neural crest cell malfunction. The role of neural crest cells in the development of the sutures are discussed in the greater details. However there are several major issues which need to be addressed before this being considered for the publication.
This review does not particularly discuss the current knowledge of molecular and genetic studies and findings of different organisms such as avian and fish models in neural crest research.
It is true that this review focuses on mouse and human studies. That said, we acknowledge that much important information using other models has contributed to the knowledge base regarding neural crest cells in development, which is now discussed. Page 3, lines 81-88.
In addition, to address this critique, the word, “mammalian” has been to introductory statements regarding what is included in the review. Page 2, lines 45 and 47.
My specific comments about this review is as follows.
Subtopic “Neural crest cells”
The origin of the neural crest cells are different in different organisms. Please be specific and include references about different mechanisms of origin of crest cells in different organisms.
Authors mentioned BMP, WNT and FGF as key cell signaling pathways of neural crest cell migration. This is bit misleading. Other than above Hh, Notch and TGF pathways signaling are important in regulation of neural crest cell fate. It is important to mention these as well.
Additional text with references has been added to includes the roles of HH, Notch and TGF superfamily in control of neural crest cell fate Page 3, lines 102-108.
The revision also includes a new figure (Fig. 1, Page 2) that is a schematic of early neural crest development.
Subtopic “Cranial neural crest cells in the development of cranial bones and coronal suture”
Line # 227-229 “coronal suture exists as an intramembranous bone growth site and a physical boundary between neural crest-derived frontal bone and paraxial mesoderm-derived parietal bone”. This information is repetitive. Check the manuscript and edit accordingly.
This sentence has been removed from this sentence to avoid repetition in the manuscript.
Subtopic “Molecular mechanisms underlying Coronal Craniosynostosis”
This section needs to be revised according to the title. What authors mean by molecular mechanisms are unclear. Here they discuss only genetic syndromes and cell signaling pathways. But needs to be include the epigenetic factors and the downstream signaling mechanisms which are critical in determining the neural crest cell fate.
The discussion about the lineage tracing studies are fine. But you can include other genetic studies such as knock down and knock in models, genome wide associated studies and studies done on other animal models.
Please note that Table 1 includes referenced cell mechanisms with established gene mutations for human syndromes that primarily involve fusion of the coronal suture. The text also includes description of numerous mouse models and downstream effects of mutations or knockouts.
That said, in response to this critique, we added additional text and subsections including a paragraph describing loss of the coronal suture due to overgrowth of the frontal bone (page 10, lines 368-382), a section on epigenetics and the craniosynostosis gene network (pages 12-13, lines 475-510) and references to online databases made available through genetic, epigenetic, GWAS and exome sequencing (Page 12, lines 498-501).
Figure revisions
Figure 1. This needs to be labeled properly as it it difficult to understand a person who does not know the anatomy of the human skull. Indicate the directions and explain the view. ex: Anterior –Posterior, name the bones.
Anterior, posterior, superior, inferior, left and right labels were added to this figure (now Fig. 2). Bones were also labeled as derived from neural crest or mesoderm. Page 5.
Figure 2.Enlarge the diagram. Label the sequence (a,b,c or 1, 11, 11)
This figure is enlarged (now Fig. 3), schematics are labeled and indicated in the figure legend. Page 7.
Figure 3. The (a) and (b) are in wrong sequence and not match with the legend. Please correct the figure or the legend.
Thank you for catching this mistake. The figure legend for this figure (now Fig. 4) is now correct. Page 14.

Reviewer 2 Report
Siismets and Hatch present a well-written and interesting overview of cranial neural crest cells and their relationship to human craniofacial anomalies. The authors have done well to provide a review of our current understanding of developmental biology and how it underpins our understanding of malformations when fundamental processes go wrong.
There are a few suggestions that I think would improve on this manuscript outlined below:
Page 2, Table 1: I suggest omitting En1 from this Table because there is currently no known associated human syndrome. The Table’s title is “…underlying human craniofacial anomalies”, so including En1 seems wrong.
Page 4, line 117: I think that the statement “Pierre Robin sequence is therefore also caused by cranial neural crest cell deficiency” could be edited to temper the statement because the underlying pathogenesis in ALL individuals with PRS is not fully resolved. Cranial NCC deficiency may be one mechanism, but I don’t think we know that it is the only mechanism in all cases. Furthermore, what does “deficiency” in this context mean? A deficiency of migration, proliferation, signalling etc?
Page 4, line 124: “…the debilitating pediatric condition…” - I suggest avoiding the use of overly dramatic or inaccurate terms such as “debilitating”. Many parents of affected children would assert that their children are not “debilitated”. Keep statements neutral and factual.
Page 5, line 129: Craniosynostosis does not “lead to brain abnormalities” - it can result in raised intracranial pressure, but the underlying brain in most children is structurally normal.
Page 9, line 309: “Crouzon patients” - I suggest using person-first language out of respect for our fellow human beings. Use “individuals with Crouzon syndrome” instead. Check the entire manuscript for examples of this e.g. “Apert syndrome patients”.
Page 10, lines 381-382: Sentence starting “Because the sole treatment…” is clumsy and difficult to understand - what exactly do the authors mean by, “risk high morbidity”? I suggest re-writing the entire sentence for clarity.
Page 11, lines 398-401: There are two sentences that have run in to each other. Suggest separating the two phrases into separate sentences.
Page 11, lines 401-419: This section is interesting and the authors allude to the potential for translation of basic science to disease-modifying therapy. I suggest that the authors loop back to this idea in the closing sentence of the paragraph (line 419).
Stylistically, some of the language used is informal and not appropriate for academic writing. Examples include:
Page 2, line 76: “Too little…” - try “insufficient”; “…too much…” - try “excess. This will set a more professional tone and reduce verbiage.
Page 4, line 76: “When defects in cranial neural crest cell arise, it commonly results in a craniofacial anomaly”. Check grammar and sentence construction. The sentence is rather simplistic and could be edited.
Page 4, line 96: “into the in” —> grammar
Author Response
We would like to thank this reviewer for their time and effort to do a thorough review of this manuscript. In response, many changes were incorporated. We hope that this reviewer agrees that the manuscript is improved.
Review 2
Siismets and Hatch present a well-written and interesting overview of cranial neural crest cells and their relationship to human craniofacial anomalies. The authors have done well to provide a review of our current understanding of developmental biology and how it underpins our understanding of malformations when fundamental processes go wrong.
There are a few suggestions that I think would improve on this manuscript outlined below:
Page 2, Table 1: I suggest omitting En1 from this Table because there is currently no known associated human syndrome. The Table’s title is “…underlying human craniofacial anomalies”, so including En1 seems wrong.
We agree with this suggestion. En1 was removed from Table 1.
Page 4, line 117: I think that the statement “Pierre Robin sequence is therefore also caused by cranial neural crest cell deficiency” could be edited to temper the statement because the underlying pathogenesis in ALL individuals with PRS is not fully resolved. Cranial NCC deficiency may be one mechanism, but I don’t think we know that it is the only mechanism in all cases. Furthermore, what does “deficiency” in this context mean? A deficiency of migration, proliferation, signaling etc?
In response to this valid critique, the statement on page 4, lines 171-173 was revised to state, “Pierre Robin sequence can therefore also be caused by diminished signaling in cranial neural crest cells leading to reduced proliferation and/or osteogenesis within the mandible.”. In addition, we added an additional reference and text to include both ERK and BMP signaling deficiencies in neural crest cells that were shown to cause Pierre Robin sequence in mice.
Page 4, line 124: “…the debilitating pediatric condition…” - I suggest avoiding the use of overly dramatic or inaccurate terms such as “debilitating”. Many parents of affected children would assert that their children are not “debilitated”. Keep statements neutral and factual.
The word “debilitating” was removed (page 4, line 179).
Page 5, line 129: Craniosynostosis does not “lead to brain abnormalities” - it can result in raised intracranial pressure, but the underlying brain in most children is structurally normal.
Some infants with severe craniosynostosis can develop chiari malformation. But, we agree overall with this critique such that “brain abnormalities” was removed from this sentence (page 5, lines 184-185).
Page 9, line 309: “Crouzon patients” - I suggest using person-first language out of respect for our fellow human beings. Use “individuals with Crouzon syndrome” instead. Check the entire manuscript for examples of this e.g. “Apert syndrome patients”.
All references to “patients” were changed in this revision to “individuals”.
Page 10, lines 381-382: Sentence starting “Because the sole treatment…” is clumsy and difficult to understand - what
exactly do the authors mean by, “risk high morbidity”? I suggest re-writing the entire sentence for clarity.
The sentence was changed to, “Individuals with craniofacial anomalies require surgical intervention. Individuals with severe phenotypes can experience high morbidity due to the need for multiple corrective surgeries throughout childhood and adolescence”. Page 13, lines 516-518.
Page 11, lines 398-401: There are two sentences that have run in to each other. Suggest separating the two phrases into separate sentences.
This sentence is now two separate sentences. Page 13, line 534.
Page 11, lines 401-419: This section is interesting and the authors allude to the potential for translation of basic science to disease-modifying therapy. I suggest that the authors loop back to this idea in the closing sentence of the paragraph (line 419).
We are appreciative of this positive comment regarding our recent preclinical study results. As suggested, the sentence, “This is but one example of how the investigation of molecular mechanisms behind specific craniofacial anomalies is essential for the development of disease-modifying therapies” is now provided at the end of this paragraph. Page 13, lines 555-557.
Stylistically, some of the language used is informal and not appropriate for academic writing. Examples include:
Page 2, line 76: “Too little…” - try “insufficient”; “…too much…” - try “excess. This will set a more professional tone and reduce verbiage.
“Too little” was replaced with “insufficient”, Page 3, line 100.
“Too much” was replaced with “excess”, Page 3, line 101.
Page 4, line 76: “When defects in cranial neural crest cell arise, it commonly results in a craniofacial anomaly”. Check grammar and sentence construction. The sentence is rather simplistic and could be edited.
This sentence now reads, “Cranial neural crest cell defects underpin many craniofacial anomalies”. Page 4, line 132.
Page 4, line 96: “into the in” —> grammar
This sentence has been corrected.
Reviewer 3 Report
This manuscript summarizes the current understanding of neural crest cells in craniofacial development, craniofacial anomalies, and the pathogenesis of coronal craniosynostosis.
The essence of this article is the detailed description of existing mutants that cause human craniofacial anomalies. Moreover, the authors focus on the cranial neural crest cell in the development of cranial bones and the coronal suture. They also summarized the underlying molecular mechanisms that cause the coronal craniosynostosis and discussed the novel approaches of prevention and/or treatment strategies for craniofacial tissue regeneration and craniosynostosis.
The article is well written in enough detail and clearly illustrated. It is an important summary of the craniofacial study field, and I fully support its publication in the Journal of Developmental Biology.
I have only one suggestion. Add a simple schematic diagram illustrating the major developmental process of neural crest, e.g., induction, EMT, and migration, as described in section 2. The diagram may also include the major signaling pathways and essential factors that regulate the processes
Author Response
We would like to thank this reviewer for their time and effort to do a thorough review of this manuscript. In response, many changes were incorporated. We hope that this reviewer agrees that the manuscript is improved.
Review 3
This manuscript summarizes the current understanding of neural crest cells in craniofacial development, craniofacial anomalies, and the pathogenesis of coronal craniosynostosis.
The essence of this article is the detailed description of existing mutants that cause human craniofacial anomalies. Moreover, the authors focus on the cranial neural crest cell in the development of cranial bones and the coronal suture. They also summarized the underlying molecular mechanisms that cause the coronal craniosynostosis and discussed the novel approaches of prevention and/or treatment strategies for craniofacial tissue regeneration and craniosynostosis.
The article is well written in enough detail and clearly illustrated. It is an important summary of the craniofacial study field, and I fully support its publication in the Journal of Developmental Biology.
I have only one suggestion. Add a simple schematic diagram illustrating the major developmental process of neural crest, e.g., induction, EMT, and migration, as described in section 2. The diagram may also include the major signaling pathways and essential factors that regulate the processes
Thank you for this suggestion. A new schematic figure has been added to illustrate early NC development (Fig. 1, Page 2).
Reviewer 4 Report
This review focuses on our current knowledge on craniofacial anomalies and specifically on craniosynostosis. The review is interesting and covers an important field of research and will be useful for students and other members of the field. While the review in general is well written, there are a few major concerns that should be addressed before publication. Also, the text contains several small sloppy mistakes, and the entire text needs to be thoroughly double checked before resubmission.
Main concerns:
- Considering that the purpose of a review article is to gather and process the latest knowledge of a given topic, too many of the references are simply too old. The reader needs to be able to trust that the authors have done their part and thoroughly read everything that has been published on the topic before then making educated choices of what to actually choose as references. All in all, 30% of the references are from 2005 or older while only 22% are from the past five years. While some original work naturally needs to be referenced despite their age, this manuscript is repeatedly referencing very old molecular biology papers in a context where much more detailed recent research exists. This is unacceptable and not only unfair to the authors of the recent work but also harmful to the field as reviews like this have a tendency to repeat and strengthen old, uncomplete knowledge that may even already have been proven wrong by newer studies.
For example, just to name a few examples, on rows 53-56, the manuscript references to work on neural crest EMT by referencing a neural crest cadherin paper from 1998, and a general EMT paper (not studying neural crest) from 2000 as well as a review on neural crest gene regulatory networks from 2015, not EMT (although several reviews have been written on the topic). The review from 2015 even references some of the more recent studies (e.g. by Rogers, Saxena, Bronner, 2013, who were the first to show that also E-cad needs to be downregulated in the NC, not only N-cad and Cad6B), and several studies on this topic have been published recently, within the last 5-10 years. In light of the more recent data, the sentence in the manuscript is not true anymore and should not be repeated in a review from 2020. Another example is the reference for the molecular mechanism behind Treacher Collins syndrome. The manuscript refers to a work from 2006 (table 1, rows 96-97) that provides a very general picture while more recent work has discovered much more detailed molecular level mechanisms (Werner et al, Nature 2015). The authors thus need to double check their references and make sure the literature review is up to date. Even just a few clearly missing central references will create an untrusting overall feeling to the reader regarding the whole article.
- As the main topic of the manuscript is craniosynostosis, the introduction could be a bit more focused in terms of providing supportive information for the main topic, and not just general details about neural crest (the current text provides a little bit details here and there that don’t really add anything relevant to understanding craniosynostosis).
- Table 1 lists mouse models for human craniofacial anomalies. The logic of what has been selected and what has been left out is unclear, there are several more known models that are not mentioned (e.g. CHARGE syndrome, ChD7, or Muenke syndrome, FgfR3 etc. to name just a few). Also, the word NEUROCHRISTOPATHY is not mentioned in the review at all and should be included, also for the search engines to find the publishd work. The authors may want to consider only listing models for craniosynostosis, which might fit the theme better, or alternatively then list literally all craniofacial malformation models that exist.
- Figure I and related text: The review would be clearer, especially to readers less familiar with the details of craniosynostosis, if this figure would show the names of the cranial bones, and also color code the origin (NC or mesoderm) of the respective bones and sutures. Having these essential details in a cartoon would make it much easier for the reader to follow the text.
- It would be interesting to have a small section about the sagittal suture as well, which is neural crest derived and how does it compare to coronal suture.
Significant minor details:
The text contains several tiny mistakes, which gives a sloppy impression (and a general feeling of mistrust towards the whole work). Below are a few examples but the authors need to thoroughly go through the text and double check everything, including the references:
- Rows 170-173: The sentence about hox genes is confusing: engrailed is not a Hox gene per se. Also, does reference 32 actually state what the sentence claims?
- Row 240 is missing a reference.
- Figure 3 figure legend does not match with the figure, please double check. A and b are reversed?, but also the grading (1-3) in the image does not match with the text description (grading 0-3).
- Row 483 “the mouse model”, (Mouse models?)
- The style of writing in some occasions can be slightly misleading. Eg. on rows 72-76, one could understand that reference 18 rules out findings in ref 17 , which may not be the case.
- As the authors name a few genes that impact a certain function but leave out other players that also have shown to play a role, please remember to state that you are only talking about some findings, not everything that is known.
Author Response
We would like to thank this reviewer for their time and effort to do a thorough review of this manuscript. In response, many changes were incorporated. We hope that this reviewer agrees that the manuscript is improved.
Review 4
This review focuses on our current knowledge on craniofacial anomalies and specifically on craniosynostosis. The review is interesting and covers an important field of research and will be useful for students and other members of the field. While the review in general is well written, there are a few major concerns that should be addressed before publication. Also, the text contains several small sloppy mistakes, and the entire text needs to be thoroughly double checked before resubmission.
Main concerns:
- Considering that the purpose of a review article is to gather and process the latest knowledge of a given topic, too many of the references are simply too old. The reader needs to be able to trust that the authors have done their part and thoroughly read everything that has been published on the topic before then making educated choices of what to actually choose as references. All in all, 30% of the references are from 2005 or older while only 22% are from the past five years. While some original work naturally needs to be referenced despite their age, this manuscript is repeatedly referencing very old molecular biology papers in a context where much more detailed recent research exists. This is unacceptable and not only unfair to the authors of the recent work but also harmful to the field as reviews like this have a tendency to repeat and strengthen old, uncomplete knowledge that may even already have been proven wrong by newer studies.
For example, just to name a few examples, on rows 53-56, the manuscript references to work on neural crest EMT by referencing a neural crest cadherin paper from 1998, and a general EMT paper (not studying neural crest) from 2000 as well as a review on neural crest gene regulatory networks from 2015, not EMT (although several reviews have been written on the topic). The review from 2015 even references some of the more recent studies (e.g. by Rogers, Saxena, Bronner, 2013, who were the first to show that also E-cad needs to be downregulated in the NC, not only N-cad and Cad6B), and several studies on this topic have been published recently, within the last 5-10 years. In light of the more recent data, the sentence in the manuscript is not true anymore and should not be repeated in a review from 2020. Another example is the reference for the molecular mechanism behind Treacher Collins syndrome. The manuscript refers to a work from 2006 (table 1, rows 96-97) that provides a very general picture while more recent work has discovered much more detailed molecular level mechanisms (Werner et al, Nature 2015). The authors thus need to double check their references and make sure the literature review is up to date. Even just a few clearly missing central references will create an untrusting overall feeling to the reader regarding the whole article.
This is an interesting critique that is certainly of importance. In this revision we made considerable effort to reference original articles that first documented fundamental genetic, phenotypic and cell/molecular mechanisms that control neural crest cells and their role in craniofacial development and craniosynostosis. This is likely why older articles are cited. That said, we agree it is essential to also include the most recent articles that provide more comprehensive analyses and/or evolution in our understanding of such developmental events.
In this revision we added additional text with and revised overall with a focus on inclusion of the most recent studies. While this did not raise the percentage of references within the last five years (because not all added references were from the last five years), we feel confident that we’ve included a comprehensive literature review that does include recent work.
- As the main topic of the manuscript is craniosynostosis, the introduction could be a bit more focused in terms of providing supportive information for the main topic, and not just general details about neural crest (the current text provides a little bit details here and there that don’t really add anything relevant to understanding craniosynostosis).
In response to this critique, we added the following text to the Introduction section. Page 1, lines 40-45.
“The coronal cranial suture is prematurely fused in many individuals with syndromic craniosynostosis, and particularly in those with mutations in FGFR2 or Twist [8, 9]. Unlike other cranial sutures, the coronal suture develops between mesoderm derived parietal and neural crest derived frontal bone rudiments, and maintains the boundary between these bone tissues of different embryonic origin during growth [10, 11]. The role of neural crest cell defects in the pathogenesis of coronal craniosynostosis is therefore of particular interest”.
- Table 1 lists mouse models for human craniofacial anomalies. The logic of what has been selected and what has been left out is unclear, there are several more known models that are not mentioned (e.g. CHARGE syndrome, ChD7, or Muenke syndrome, FgfR3 etc. to name just a few). Also, the word NEUROCHRISTOPATHY is not mentioned in the review at all and should be included, also for the search engines to find the published work. The authors may want to consider only listing models for craniosynostosis, which might fit the theme better, or alternatively then list literally all craniofacial malformation models that exist.
In response to this critique, Table 1 is now limited to human syndromes that include coronal suture fusion as a primary phenotype. FGFR2 mutations causing bent bone dysplasia, FGFR3 mutations causing Muenke syndrome and TCF12 mutations causing “mild” Saethre-Chotzen syndrome are now also included. Page 9.
Thank you for pointing out that we did not use the term neurocristopathy anywhere in the review. In response to this suggestion, additional text has been added and reference to a comprehensive review of neurocristopathies is provided. Pages 2-3, lines 122-131.
- Figure I and related text: The review would be clearer, especially to readers less familiar with the details of craniosynostosis, if this figure would show the names of the cranial bones, and also color code the origin (NC or mesoderm) of the respective bones and sutures. Having these essential details in a cartoon would make it much easier for the reader to follow the text.
In response to this and to reviewer 1 critique, this figure (now Fig. 2) and its legend were revised to include bone labels and embryonic origin. Page 5.
- It would be interesting to have a small section about the sagittal suture as well, which is neural crest derived and how does it compare to coronal suture.
Development of the sagittal suture is now included. Page 8, lines 302-307.
Significant minor details:
The text contains several tiny mistakes, which gives a sloppy impression (and a general feeling of mistrust towards the whole work). Below are a few examples but the authors need to thoroughly go through the text and double check everything, including the references:
While we attempted to provide an entirely accurate review, we appreciate the errors that this reviewer pointed out. All attempts have been made in this revision to avoid such mistakes.
- Rows 170-173: The sentence about hox genes is confusing: engrailed is not a Hox gene per se. Also, does reference 32 actually state what the sentence claims?
Msx and Dlx genes are now referred to as “homeobox containing transcription factors”. Page 6, lines 238-239.
En1 is now correctly stated as a protein that is “homeobox domain containing”. Page 6, line 271.
Reference 32 has been replaced by other original studies of cleft palate heritability (references 62,63), Page 4, line 152. We also now refer to a recent comprehensive review of cleft palate, that readers may find helpful (reference 61). Page 4, lines 151-152.
- Row 240 is missing a reference.
References are now provided. Page 11, line 426.
- Figure 3 figure legend does not match with the figure, please double check. A and b are reversed?, but also the grading (1-3) in the image does not match with the text description (grading 0-3).
This figure legend has been corrected (now Fig. 4). Page 14, lines 559-570.
- Row 483 “the mouse model”, (Mouse models?)
This phrase has been edited to read, “Advances in mouse genetic technologies for generation of relevant mouse models”. Page 15, lines 616-617.
- The style of writing in some occasions can be slightly misleading. Eg. on rows 72-76, one could understand that reference 18 rules out findings in ref 17 , which may not be the case.
These sentences are now modified: “FGF8 can promote FGF2 activity and neural crest cell chemotaxis [31,32]. More recent studies indicate that FGF8 can also promote neural crest cell survival in the first pharyngeal arch as evidenced by data showing that FGF8 inactivation caused massive NCC apoptosis in the first pharyngeal arch [33]. Page 3, lines 97-99.
- As the authors name a few genes that impact a certain function but leave out other players that also have shown to play a role, please remember to state that you are only talking about some findings, not everything that is known.
This is an excellent point. This revision includes reference to this fact via the statement, “There are currently 3,164 genes in Online Mendelian Inheritance in Man® listed under the search term “craniosynostosis causing”, indicating that that we are perhaps at the tip of the iceberg in terms of understanding the craniosynostosis genetic network [192]”. Page 13, lines 511-514.